

# How large is the design space for stratospheric aerosol geoengineering?

Yan Zhang[1], Douglas G. MacMartin[1], Daniele Visioni[1], and Ben Kravitz[2,3]

[1]Sibley School of Mechanical and Aerospace Engineering, Cornell University, Ithaca, NY, USA
[2]Department of Earth and Atmospheric Science, Indiana University, Bloomington, IN, USA
[3]Atmospheric Sciences and Global Change Division, Pacific Northwest National Laboratory, Richland, WA, USA

**Correspondence:** Yan Zhang (yz2545@cornell.edu)

**Abstract.** Stratospheric aerosol injection (SAI), as a possible supplement to emission reduction, has the potential to reduce some of the risks associated with climate change. Adding aerosols to the lower stratosphere results in global cooling. However, different choices for the aerosol injection latitude(s) and season(s) have been shown to lead to significant differences in regional surface climate, introducing a design aspect to SAI. Past research has shown that there are at least three independent degrees of

freedom (DOF) that can be used to simultaneously manage three different climate goals. Knowing how many more DOFs there are, and thus how many independent climate goals can be simultaneously managed, is essential to understanding fundamental limits of how well SAI might compensate for anthropogenic climate change, and evaluating any underlying trade-offs between different climate goals. Here we quantify the number of meaningfully-independent DOFs of the SAI design space. This number of meaningfully-independent DOFs depends on both the amount of cooling and the climate variables used for quantifying the

changes in surface climate. At low levels of global cooling, only a small set of injection choices yield detectably different surface climate responses. For a cooling level of 1-1.5°C, we find that there are likely between 6 and 8 meaningfully-independent DOFs. This narrows down the range of available DOF and also reveals new opportunities for exploring alternate SAI designs with different distributions of climate impacts.

## 1   Introduction

Reducing emissions of $CO_2$ and other greenhouse gases (GHG) may not be enough by itself to avoid significant risks associated with climate change. As a supplement to emission reduction, climate interventions such as stratospheric aerosol injection (SAI) may be able to reduce some of these risks. SAI involves adding aerosols or their precursors to the lower stratosphere, which would increase the stratospheric aerosol optical depth (AOD); as a result, more solar radiation would be reflected away before reaching the surface. Most climate model simulations inject $SO_2$, which results in increased sulfate aerosols. While injecting

aerosols (or a precursor gas such as $SO_2$) into the stratosphere can offset the change in global mean temperature, the resulting climate would not be the same as the climate with the same temperature but without either the excess atmospheric $CO_2$ or SAI. These residual changes depend on injection choices that could be made. As suggested in previous research, injecting aerosols at different latitudes, altitudes, and seasons would result in different spatiotemporal patterns of AOD, which in turn would lead to different regional surface climate responses (MacMartin et al., 2017; Tilmes et al., 2017, 2018; Dai et al., 2018; Kravitz





et al., 2019; Visioni et al., 2019, 2020c; Lee et al., 2020a, 2021). Understanding the global and spatiotemporal impacts of SAI
and even the governance challenges requires that we not treat SAI as one single strategy, but rather understand the range of
outcomes across different strategies, the fundamental limits of how well SAI can compensate for GHG-driven climate change,
and any underlying trade-offs among SAI strategies.

Choosing where and when to inject aerosols can be thought of as a design problem (Ban-Weiss and Caldeira, 2010; Kravitz

et al., 2016; MacMartin and Kravitz, 2019); for a given choice of global cooling, the design space describes the range of
all possible such injection strategies. Some strategies produce very different surface climate responses, while others can be
relatively similar. These climate responses can be quantified by different metrics, such as surface air temperature, precipitation,
and Arctic sea ice. We use the term "degrees of freedom" (DOF) to describe how many independent injection choices there
are in the design space. The number of independent injection choices is equivalent to the number of independent climate goals

that can be managed by SAI simultaneously. Most studies have only explored a single degree of freedom: injecting $SO_2$ at one
location (often the equator) either at a fixed rate or to meet one climate objective (often global mean temperature (T0)) (e.g.,
Robock et al., 2008; Rasch et al., 2008; Kravitz et al., 2011, 2015). Kravitz et al. (2016, 2017) demonstrated a strategy in which
three DOFs were used to manage three temperature goals: T0, interhemispheric temperature gradient (T1), and equator-to-pole
temperature gradient (T2); the same strategy was then used in The Geoengineering Large Ensemble Project (GLENS) (Tilmes

et al., 2018). Additional studies have explored variations on these DOFs, such as Visioni et al. (2020c), who injected $SO_2$ in
only one season to meet the same set of climate goals, or Lee et al. (2020a), who used the same set of DOFs as in Tilmes et al.
(2018) to meet different sets of climate goals, including T0, the latitude of the Intertropical Convergence Zone (ITCZ), the
amount of Arctic September Sea Ice (SSI), and global mean precipitation (P0). Higher-latitude injections in different seasons
have been shown to have different efficacies in preserving SSI; e.g., spring-only injection at 60° N restores twice the amount of

SSI compared to annually-constant injection at that latitude (Lee et al., 2021). A key open question from these design studies is
how many other strategies are unexplored (e.g., MacMartin and Kravitz, 2019); in other words, how many independent degrees
of freedom are there?

In this study, we estimate the number of DOF of the design space for SAI. Knowing how many DOFs there are in the design
space quantifies the number of independent climate goals that can be managed simultaneously by a SAI strategy. In order to be

managed simultaneously, those independent climate goals cannot conflict. (For example, T0 and P0 are conflicting and cannot
be managed simultaneously; see e.g., Bala et al., 2008, Tilmes et al., 2013, and Lee et al., 2020a.) Knowing the number of
DOF also helps understand the full range of possible climate outcomes and what climate outcomes cannot be achieved by SAI
strategies. In this study, we focus only on $SO_2$ injections, and evaluate the range only in one model. However, the results will
depend primarily on the constraints imposed by stratospheric circulation and the lifetime of the aerosols in the stratosphere

(Tilmes et al., 2017; MacMartin et al., 2017; Dai et al., 2018); as such, many of the conclusions can be expected to be applicable
regardless of aerosol choice.

The aerosols will primarily stay in the same hemisphere where they are injected and be transported mostly poleward by
the stratospheric Brewer-Dobson circulation (Tilmes et al., 2017; MacMartin et al., 2017). Thus, injecting in one hemisphere
preferentially increases AOD in that hemisphere; injecting further poleward increases the AOD burden further poleward. In-





jecting above the equator produces an AOD peak in the tropics (Kravitz et al., 2019). More generally, different choices of injection latitude, altitude and season lead to different spatiotemporal AOD patterns as a result of the seasonally-varying stratospheric circulation. However, not all choices contribute the same level of "uniqueness" (MacMartin et al., 2017; Visioni et al., 2019, 2020c). For example, injecting at the equator would produce very different patterns of AOD compared to injecting at 30° N, but the patterns of AOD arising from injecting at 31° N should not be expected to be very different from injecting at

30° N. As more choices of injection latitude are considered, there exists a diminishing return on the "uniqueness" contributed by additional choices of injection latitude. That leads to the question of how many meaningfully-independent patterns of AOD are possible given the constraints imposed by stratospheric circulation.

There are two distinct steps in the analysis herein. The first step is to consider how different the spatiotemporal AOD patterns are for different injection choices. And second, to know whether the AOD patterns from two different injection choices are

sufficiently similar to treat them as effectively equivalent, or sufficiently distinct to treat them as two separate DOFs, one needs to relate how similar or dissimilar the patterns of AOD are to how similar or dissimilar the resulting climate responses are. Identifying the number of DOF only needs to consider injections that produce meaningfully different climates; herein we define "meaningfully different" based on the ability to detect the difference in climate after 20 years, given natural variability – and this threshold clearly depends on the choice of climate variables to be considered and the amount of cooling desired.

For example, injecting aerosols at 30° N only in the summer or only in the fall would yield different patterns of AOD. The difference in resulting climate could be distinguishable against the background climate variability if the desired amount of cooling is high. However, by reducing the level of cooling, the difference would become indistinguishable.

The next section describes the climate model and simulations used. Section 3 assesses the differences in spatiotemporal AOD patterns from 29 different injection choices, sampling different latitudes and seasons of injection, and quantifies the size

of design spaces with different numbers of DOF using a metric based on the angle between different patterns of AOD. Section 4 identifies a relationship between how similar or dissimilar AOD patterns are and how similar or dissimilar the corresponding climate responses are, using existing simulations that were conducted with various different choices for climate goals and/or different DOFs. Section 5 then quantifies how large the difference in climates needs to be in order to be meaningfully different at different levels of cooling. Finally, we combine these pieces of analysis in Section 6 to show that for a cooling level of

1-1.5°C, for example, there are between 6 and 8 DOFs.

## 2   Model and Simulations

All simulations in this study were conducted using the Community Earth System Model version 1 with the Whole Atmosphere Community Climate Model as the atmospheric component, CESM1(WACCM). CESM1(WACCM) is a fully coupled Earth System model which includes atmosphere, ocean, land, and sea ice components (Mills et al., 2017). The model has a horizontal

resolution of 0.95° latitude by 1.25° longitude, with 70 vertical levels that extend from the earth surface to 140 km in altitude, and stratospheric aerosols have been shown to reasonably match observations after the Mt. Pinatubo eruption (Mills et al., 2017). With the exception of a few cases noted below, we use existing output from previous simulations for our analysis.



To assess the range of possible spatiotemporal patterns of AOD arising from different injection choices, we sample 29
possible choices in the AOD design space, including injections at low and middle latitudes as described by Visioni et al. (2019),
and high latitude injections as described by Lee et al. (2021); the set of 29 possible choices is illustrated in Fig. 1. Visioni et al.
(2019) include injections at 5 different latitudes: 30° N, 15° N, equator, 15° S, and 30° S, as in Tilmes et al. (2017). For each
latitude, injections are simulated both annually-constant, and restricted to each season: December-January-February (DJF),
March-April-May (MAM), June-July-August (JJA), and September-October-November (SON). In each simulation, $6\ \mathrm{Tg\ yr^{-1}}$
of $SO_2$ are injected into the lower stratosphere, approximately 6-7 km above the tropopause at 180° E (about 25 km for
equator and 15° N/S, 23 km for 30° N/S, 16 km for 45° N/S, and 15 km for 60° N/S). Simulations were conducted for 5
years (2040-2044), which is sufficient for estimating the steady-state AOD pattern (Visioni et al., 2019), though of course not
for estimating the climate response to this forcing. The high latitude injections included here are not exactly the same as those
described in Lee et al. (2021), which have a higher injection rate of $12\ \mathrm{Tg\ yr^{-1}}$ and only consider injection at 60° N and
further poleward. To be consistent with simulations performed by Visioni et al. (2019), we conducted additional simulations
of spring (MAM or SON) injection at 45° N, 45° S, 60° N, and 60° S for 5 years from 2040 to 2044 with an injection rate
of $6\ \mathrm{Tg\ yr^{-1}}$, to complete the sample set. Following Lee et al. (2021), the other seasons of injection at high latitude are not
expected to be particularly effective. Figure 2 shows the spatiotemporal patterns of AOD in each of these four spring injections
at high latitudes (see Visioni et al., 2019 for the remaining cases). This gives us a total of 29 different injection cases, and
associated spatiotemporal patterns of AOD. In Section 3, we rank the 29 injection cases based on the uniqueness of their AOD
patterns, which are then used to identify the number of meaningfully-independent injection choices. In Section 7, we show that
the uniqueness of stratospheric AOD patterns does not depend on the altitude of injection, using simulation data from Tilmes
et al. (2017) that include $SO_2$ injection either 1 km above the tropopause or 6-7 km above the tropopause.

In addition to these shorter simulations that we use to assess the range of possible spatiotemporal AOD patterns from
different injection choices, five sets of solar geoengineering simulations from existing studies (Tilmes et al., 2018; Kravitz
et al., 2019; Visioni et al., 2020c; Lee et al., 2020a) are used to analyze the connection between the patterns of stratospheric
AOD and surface climate responses (see Table 1). These 5 sets were all performed in CESM1(WACCM) with RCP8.5 as the
background warming scenario, and used a feedback algorithm (Kravitz et al., 2017) to adjust $SO_2$ injection rates to maintain
one or more climate objectives. Each simulation takes the 20-year average of annual mean temperature from 2010-2029 in the
Representative Concentration Pathway 8.5 (RCP 8.5) emissions scenario as the target value for T0. Maintaining T0 constant at
2010-2029 average results in 4°C of global mean cooling in each of these simulations by 2070-2089. The equatorial case adjusts
the single $SO_2$ injection rate to meet T0. The other simulations adjust $SO_2$ injection rates at multiple latitudes to simultaneously
meet T0 and additional climate objectives (see Table 1); these additional objectives include T1, T2, ITCZ (using the centroid
of precipitation between 20° S and 20° N as a proxy), and SSI. Table 1 lists the injection seasons and latitudes, the number of
ensemble members, and the design objectives of the five sets of simulations.

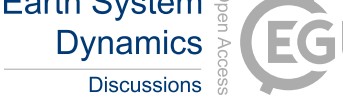

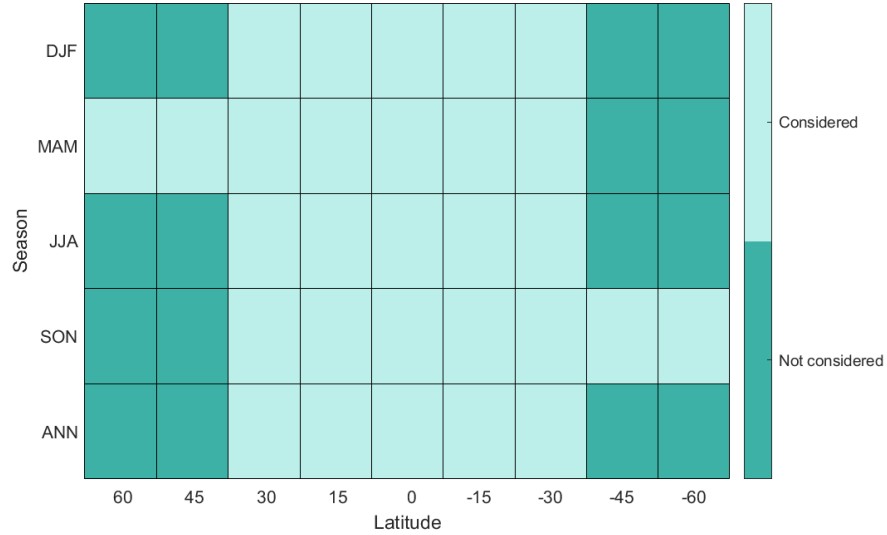

**Figure 1.** The 29 injection choices that we considered in our analysis for AOD patterns are shown in light green. Vertical axis shows the injection season of each injection choice, either injecting in only one season (DJF, MAM, JJA, or SON) or constantly throughout the year (ANN). Horizontal axis shows the injection latitude, from left to right are 60° N, 45° N, 30° N, 15° N, equator, 15° S, 30° S, 45° S, and 60° S.

**Table 1.** Injection design and outcomes of the 5 existing SAI simulations analyzed in this study.

| Name of simulation | Injection latitude | Injection season | Number of ensemble members | Objectives | Reference |
|---|---|---|---|---|---|
| GLENS | 30° N, 15° N, 15° S, 30° N | annually-constant | 21 | T0, T1, T2 | Tilmes et al. (2018) |
| iSpring | 30°N, 15° N, 15° S, 30° S | MAM at 30° N and 15° N; SON at 15° S and 30° S | 3 | T0, T1, T2 | Visioni et al. (2020c) |
| iAutumn | 30° N, 15° N, 15° S, 30° S | SON at 30° N and 15° N; MAM at 15° S and 30° S | 3 | T0, T1, T2 | Visioni et al. (2020c) |
| Equatorial | Equator | annually-constant | 3 | T0 | Kravitz et al. (2019) |
| PREC (1[st] simulation in Lee et al., 2020a) | 30° N, 15° N, 15° S, 30° N | annually-constant | 1 | T0, ITCZ, SSI | Lee et al. (2020a) |





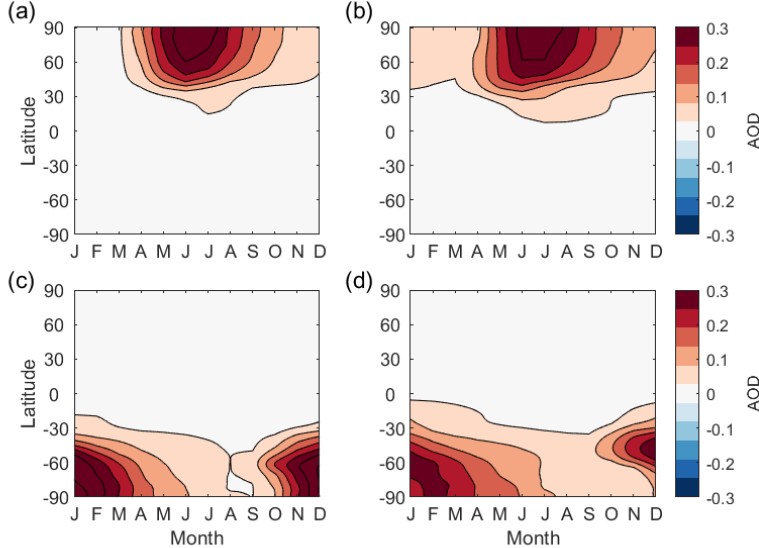

**Figure 2.** Spatiotemporal AOD patterns of spring injections at (a) 60° N, (b) 45° N, (c) 60° S, and (d) 45° S. The AOD patterns of spring injections in the same hemisphere are similar to each other, while the injections in the opposite hemispheres produce very different AOD patterns.

## 3 Diminishing Returns on the Number of Degrees of Freedom

In this section, we consider 29 different injection choices, sampling from different latitudes and seasons of injection, as well as 3 additional cases that we use to verify that the set of 29 is sufficiently complete. The AOD pattern from a given injection choice (a given latitude and season of injection) is largely determined by the stratospheric circulation and aerosol lifetime, which constrains what spatiotemporal patterns are achievable (Tilmes et al., 2017; MacMartin et al., 2017). The AOD resulting from any particular choice of latitude and season can be approximated by a linear combination of other choices. Our goal in this paper is to determine how many distinct injection choices are needed to adequately approximate all of the possible AOD patterns. What constitutes "adequacy" is determined in subsequent sections.

To describe any pattern of AOD, we consider the zonal-mean pattern as a function of both latitude and time of year. In order to treat these two dimensions consistently and yield AOD patterns independent of our sampling resolution in each dimension, we weight the monthly-mean zonal mean AOD at each latitude and month by the corresponding incoming solar energy (petajoules) at the top of the atmosphere (TOA). We then represent the weighted spatiotemporal AOD pattern from each injection choice as a vector $a$, the length of which is equal to the number of latitudes times the number of months, $\ell = \ell_{\text{lat}} \times \ell_{\text{month}}$. One way of quantifying how similar or dissimilar two patterns of AOD are is to consider the angle between their vector representations, $\theta_{AOD}$. This implicitly assumes that the patterns of AOD are sufficiently linear for 1-4°C of cooling, although nonlinearities will become increasingly important at higher forcing levels (MacMartin et al., 2017; Visioni et al.,





2020b). Thus, the angle between two vectors $a_i$ and $a_j$ that represent the AOD patterns of the same injection choice with different injection rates is negligible, while only the magnitudes of those vectors are different. Two AOD patterns that are different only in magnitude can thus be matched by adjusting injection rates, and thus are not considered as meaningfully different. Therefore, the angle between two vectors $a_i$ and $a_j$ describes how meaningfully different these two AOD patterns

are. We can illustrate this using the AOD patterns shown in Fig. 2. The AOD patterns of injection at 60° N and 60° S are very dissimilar; the angle between the vector representations of these two AOD patterns is 84°. By contrast, the AOD patterns of injection at 60° N and 45° N are similar; the angle between these two AOD patterns is only 12°.

With the vector representation explained above, our goal is to select a subset from the set of 29 injection choices such that any possible AOD pattern can be adequately represented by a linear combination of injection choices from this subset.

Determining the dimension of that set necessary to meet this goal is equivalent to determining the number of DOF of SAI.

First, we need to verify that our set of 29 injection choices sufficiently describes all of the possible AOD patterns of other injection choices that we have not simulated. To do so, we choose 3 additional verification cases, which are annual injections at 7.5° N, 22.5° N, and 37.5° N, and quantify how well each of these can be represented by a linear combination of the 29 injection choices.

Mathematically, the linear combination that is most similar to the simulated pattern of AOD is the projection of its vector representation onto the space formed by the 29 injection choices. Solving the best approximation of the pattern of AOD can be formed as a constrained linear least-square problem of finding the projection onto the set of 29 injection choices:

$$\underset{\hat{x}}{\arg\min} ||\hat{d}(\hat{x}) - d|| \tag{1}$$

$$\text{sbj} \quad \text{to} \quad \hat{d}(\hat{x}) = Q_{29}\hat{x} \tag{2}$$

$$\hat{x}_i \geq 0, \quad i = 1, ..., 29 \tag{3}$$

where $d$ is the vector representation of the AOD pattern of each verification case, which is obtained from CESM1(WACCM) simulation, $\hat{d}$ is the best approximation of $d$, $Q_{29}$ is the set of vector representations of the 29 injection choices, and $\hat{x}$ is the vector of best approximating linear coefficients. All linear coefficients $\hat{x}_i$ are constrained to be non-negative numbers, as injection rates cannot be negative.

By calculating the angle between the vector representation of the simulated AOD pattern and the vector representation of the approximated AOD pattern, we can assess how similar the simulated and approximated AOD patterns are. For annually-constant injections at 7.5° N, 22.5° N, and 37.5° N, the angles between simulated and approximated AOD patterns are 7.6° , 5.9° , and 6.1° , respectively. The AOD from injection at 7.5° N is roughly a linear combination of injections at the equator and 15° N, with a little over half of $SO_2$ injected at 15° N. Similarly, the AOD from injection at 22.5° N is roughly a linear

combination of injections at 15° N and 30° N, and the AOD from injection at 37.5° N is roughly a linear combination of injections at 30° N and 45° N. The simulated and approximated spatiotemporal AOD patterns of these three verification cases are shown in Fig. 3. Although we do not verify for an injection on the Southern Hemisphere, we expect to see a similar linear combination of injections on the same hemisphere at some latitudes lower and higher than its injecting latitude. Based on this





**Figure 3.** Comparison of simulated AOD patterns and the best approximation to these AOD patterns obtained from a linear combination of other injection choices within the set of 29 cases considered here. From top to bottom are the spatiotemporal AOD patterns for injections at 7.5° N, 22.5° N, and 37.5° N, respectively. In each horizontal panel, plots from left to right are the AOD pattern obtained from CESM1(WACCM) simulations, the AOD pattern approximated using the set of 29 injection cases, and the difference between these. The angles between the simulated AOD pattern and the AOD pattern approximated using the set of 29 cases are 7.6° , 5.9° , and 6.1° , respectively.

comparison, the approximated AOD patterns are adequately similar to the simulated AOD patterns, as long as the threshold for

an "adequate" approximation of the AOD is larger than 7.6° (which it will be here, as shown in subsequent sections). Allowing this small difference between the approximated and simulated AOD patterns, the set of 29 injection choices thus adequately describes other possible AOD patterns that we have not simulated.


With the set of 29 choices of injection locations/times, we can evaluate a wide range of possible selections of injection choices. Sets with different numbers of injection choices as well as different selections of the same number of injection choices

will do a better or worse job at spanning the space of possible AOD patterns. One way to quantify how well the overall space of possible AOD patterns can be approximated by a particular subset of $n$ injection choices is to compute the maximum angle, $\theta_{max}$, that can be formed between the subset of $n$ choices and any other injection choices that are not selected. That is, how well can the AOD pattern from any other choice be represented by a linear combination of the $n$ elements in the subset? The smaller $\theta_{max}$ is, the better the AOD pattern from any other possible injection choice can be equivalently obtained by only

choosing injections from the subset of selected injection choices. This provides a way to quantify how well the overall space is approximated by a particular subset of $n$ injection choices. By optimizing over $\theta_{max}$, we can determine both what the "best" subset is of any given dimension (number of DOF or injection choices), and equally important, how much error there would be in trying to capture any achievable AOD pattern with only a relatively few different injection choices.

In choosing subsets, we enforce hemispheric symmetry such that if an injection choice in one hemisphere is included, then

the corresponding choice on the opposite hemisphere is also included (e.g., MAM in the Northern Hemisphere and SON in the Southern Hemisphere). While the seasonal circulation patterns are not exactly symmetric between the hemispheres, they are sufficiently similar that this is a reasonable simplification that reduces the number of sets to search over. For injections at the equator, we similarly either include or don't include opposite seasons (e.g., DJF and JJA, or MAM and SON). With hemispheric symmetry, the only way to have a set with an odd number of DOF is to include annually-constant equatorial

injection; we revisit this case in the discussion.

Mathematically, the steps above can be described as follows. First, for each subset $Q_j$ of $n$ injection choices, we identify the maximum angle that can be formed between that subset and any other injection choices in the set of 29 that are not selected:

$$\theta_{max}(Q_j) = \max_{1 \leq i \leq k} \angle(\boldsymbol{a}_i, Q_j), \quad \boldsymbol{a}_i \in [Q_{29} \setminus Q_j] \tag{4}$$

where $k = 29 - n$, the total number of injection choices that are not selected by the set $Q_j$.

Taking $n = 4$ as an example, we can identify all possible combinations of 4 injection choices from the set of 29. With the enforced constraint of hemispheric symmetry, there are a total of 91 different combinations. For each of these possible combinations, we calculate the angle formed between each of the unselected injection choices and the selected set of 4 and find the maximum angle. For illustration, Fig. 4 shows an example (sub-optimal) set of 4 injection choices: summer injection at 30° N, 15° N, 15° S, and 30° S, and lists the angles formed between each of the unselected injection choices and the example set

of 4.

Among all possible subsets of $n$ injection choices ($Q_j \in Q$), we identify which subset has the smallest maximum angle and denote the "best" subset that minimizes $\theta_{max}$ as $Q^*(n)$:

$$Q^*(n) = \underset{Q_j \in Q}{\arg\min}\, \theta_{max}(Q_j) \tag{5}$$

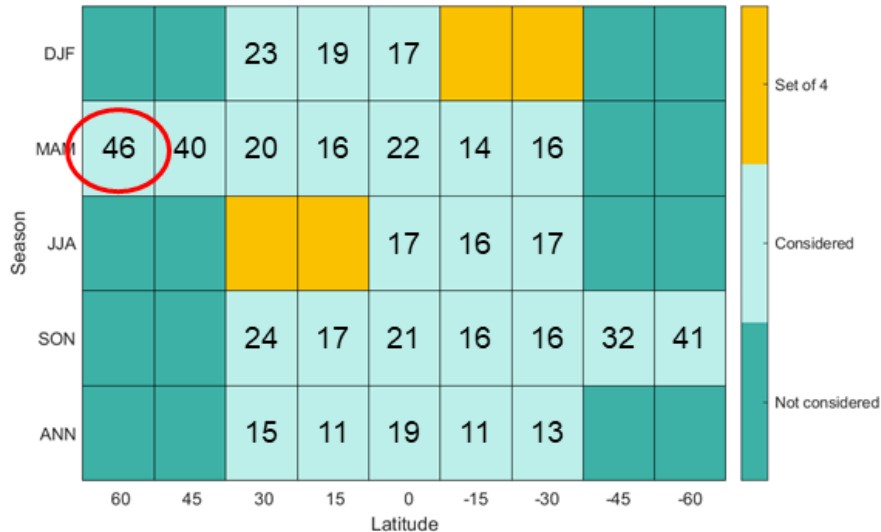

**Figure 4.** Angles (in degrees) between each unselected injection choice and a set of 4 injection choices (in orange): summer injection at $30°$ N, $15°$ N, $15°$ S, and $30°$ S. The x-axis is the injection latitude and y-axis is the injection season. The maximum angle is $46°$ (highlighted by a red circle), formed between spring injection at $60°$ N and this set of 4.

$Q^*(n)$ is the subset of size $n$ that best approximates any achievable pattern of AOD. The angle $\theta_{max}$ of the "best" set of $n$ is denoted as $\theta^*(n)$.

Still using $n = 4$ as an example, we calculate the maximum angle for all possible combinations of 4 injection choices. The set of spring injections at $45°$ N and $45°$ S and autumn injections at $15°$ N and $15°$ S has the smallest maximum angle, which is $26°$ (Fig. 5).

For each possible value of $n$, we find the "best" set and corresponding angle $\theta^*(n)$, plotted in Fig. 6. Strictly speaking, because we only sampled 29 possible injection choices (out of an infinite theoretical space), $\theta^*(n)$ is simply our best estimate for the maximum angle between a subspace of $n$ DOFs and any possible AOD pattern of injection choices that does not fall into this subspace.

## 4 Comparing AOD and Surface Climate

Figure 6 shows that there are diminishing marginal returns for how many degrees of freedom are included. However, the analysis in Section 3 does not indicate what degree of approximation is sufficient. The next step in our analysis is to evaluate the relationship between how similar or dissimilar two AOD patterns are, and how similar or dissimilar the corresponding surface climate responses are. This relationship is crucial for determining a threshold for whether two patterns of AOD from



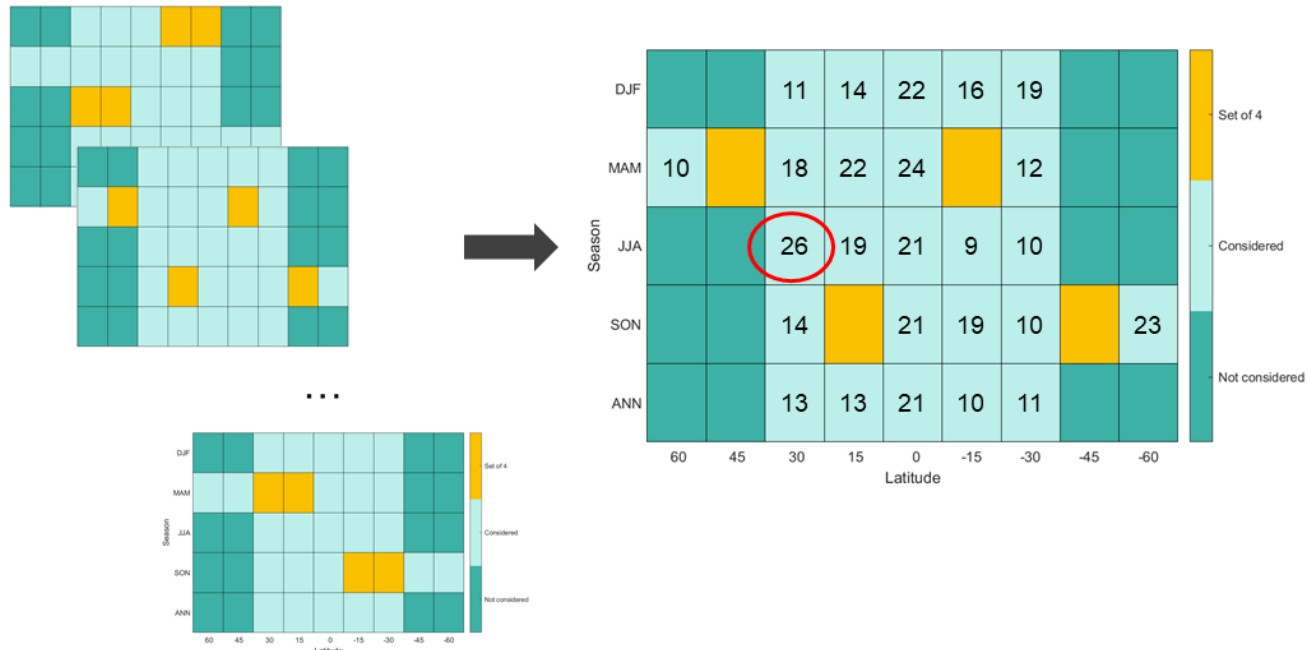

**Figure 5.** A schematic diagram showing how to find the smallest maximum angle $\theta^*(4)$ for a set of 4 injection choices. We first look through all possible combinations of 4 injection choices and calculate the maximum angle $\theta_{max}$, and then identify which set has the smallest value of $\theta_{max}$. The "best" set of 4 includes spring injections at $45°$ N and $45°$ S and autumn injections at $15°$ N and $15°$ S. The maximum angle for this "best" set of 4 is $26°$ (highlighted by a red circle), formed between this set and summer injection at $30°$ N; this is much smaller than $46°$, the maximum angle found in the example in Fig. 4.

different injection choices are sufficiently different to count as two independent degrees of freedom, or sufficiently similar to be effectively the same choice.

To estimate this relationship, we consider the different strategies described in Table 1. Each of these uses either different injection choices or has different climate goals, leading to different patterns of AOD and corresponding different surface climate responses. By comparing the difference in AOD patterns and surface climate responses, we can derive a function that describes how the similarity in surface climate responses relates to the similarity in the AOD patterns that they arise from.

        The AOD pattern and corresponding surface climate responses are obtained by averaging over all available ensemble mem-
bers for each strategy in Table 1, and taking the difference between the 2070-2089 average, and the 2010-2029 average in the RCP8.5 emissions scenario. As explained in Section 3, the monthly-mean zonal-mean AOD is weighted by the TOA incoming solar energy as a function of latitude and time of year. For the climate response, this establishes the changes from a climate with neither SAI nor increased greenhouse gases to a climate with both. To evaluate how the surface climate varies for different strategies, herein we only consider annual-mean surface air temperature and precipitation; this assumes that if these two



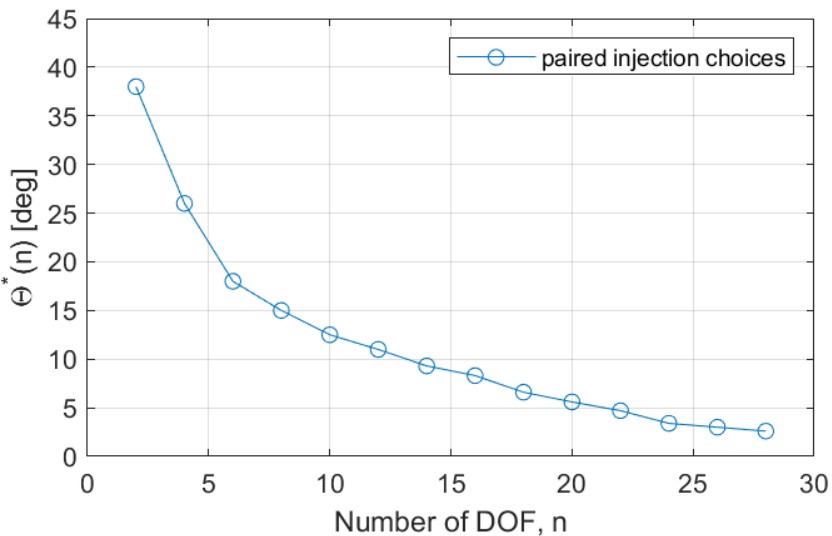

**Figure 6.** The maximum angle $\theta^*(n)$ formed between a subspace of $n$ DOFs and any other injection choices outside this subspace decreases as the number of DOF $n$ increases.

variables are similar, then changes in other surface climate variables, such as precipitation minus evaporation (P-E), will also be sufficiently similar, and also ignores shifts in the seasonal cycle (Jiang et al., 2019) as these tend to be smaller than the annual-mean changes.

To estimate how large a change in the spatiotemporal pattern of AOD is needed to obtain a detectably different pattern of surface climate response, we consider detectability over a 20-year period. Therefore, we normalize the surface temperature and
precipitation changes by the variability in 20-year averages, calculated from the across-ensemble variability from 2010-2029 in RCP8.5 emissions scenario, where 21 ensemble members are available. If the variability were uncorrelated from year to year, this value would simply be a factor of $\sqrt{20}$ smaller than the interannual variability; this would be a reasonable approximation for precipitation but not for temperature. Normalizing by variability also allows temperature and precipitation changes to be compared in consistent units (Ricke et al., 2010).

To analyze the differences in AOD and surface climate for different strategies, we define the AOD space, temperature space and precipitation space. In Section 3, we define the vector representation of AOD patterns as $\boldsymbol{a}$. $\boldsymbol{a}$ represents an achievable spatiotemporal AOD pattern arising from a possible injection choice. The AOD space $\mathcal{A}$ is a $\ell$-dimensional space, $\mathcal{A} \subset \mathbb{R}^\ell$, that includes all possible values of $\boldsymbol{a}$. Similarly, the temperature space $\mathcal{T}$ and the precipitation space $\mathcal{P}$ are both $m$-dimensional spaces, $\mathcal{T} \subset \mathbb{R}^m$ and $\mathcal{P} \subset \mathbb{R}^m$, where $m$ is equal to the number of latitudes times the number of longitudes, $m = m_{\mathrm{lat}} \times m_{\mathrm{lon}}$.
Any vector $\boldsymbol{T}$ in $\mathcal{T}$ represents a possible surface air temperature response to a possible injection choice, and any vector $\boldsymbol{P}$ in $\mathcal{P}$ represents a possible precipitation response to a possible injection choice:





$$\boldsymbol{T} = [T_1, T_2, \ldots, T_m]^T, \quad \boldsymbol{T} \in \mathcal{T} \tag{6}$$

$$\boldsymbol{P} = [P_1, P_2, \ldots, P_m]^T, \quad \boldsymbol{P} \in \mathcal{P} \tag{7}$$

where $T_1, \ldots, T_m$ are temperature responses, and $P_1, \ldots, P_m$ are precipitation responses, both in dimensionless units of stan-
dard deviations.

     In the AOD space, temperature space and precipitation space defined above, we evaluate the differences between each possible pair of the five SAI strategies described in Table 1, i.e., 10 pairwise comparisons. As in Section 3, the difference between AOD patterns for different pairs of strategies is evaluated by computing the angle between them, $\theta_{AOD}$. For temperature and precipitation, the difference between two strategies is evaluated by computing the temperature distance, $d_t$, or the precipitation
distance, $d_p$. The temperature distance is defined as the area-weighted $L^2$-norm (root-mean-square) of the difference between the two vector representations of surface air temperature responses, to count all areas on the Earth equally. Similarly, the precipitation distance is defined as the area-weighted $L^2$-norm (root-mean-square) of the difference between the two vector representations of precipitation responses. Among the 10 pairwise comparisons, GLENS and EQ have the largest temperature distance, though it's still an order of magnitude smaller than the temperature distance between GLENS and the projected 20-
year average (2070-2089) climate response under RCP8.5 (Fig. 7(a) and (b)). The precipitation distance between GLENS and EQ is also smaller than that between GLENS and RCP8.5 (Fig. 7(c) and (d)). Among all 10 pairwise comparisons, the temperature distances are always larger than the corresponding precipitation distance. That is the changes in temperature, compared to variability, are larger than the changes in precipitation. To conclude, a small angle between the vector representations of AOD patterns indicates the two compared SAI strategies yield similar AOD patterns, and a small value of $d_t$ or $d_p$ indicates
that the two compared SAI strategies have similar surface air temperature responses or precipitation responses. Likewise, a larger AOD angle, temperature distance, or precipitation distance implies less similar AOD patterns, surface air temperature or precipitation.





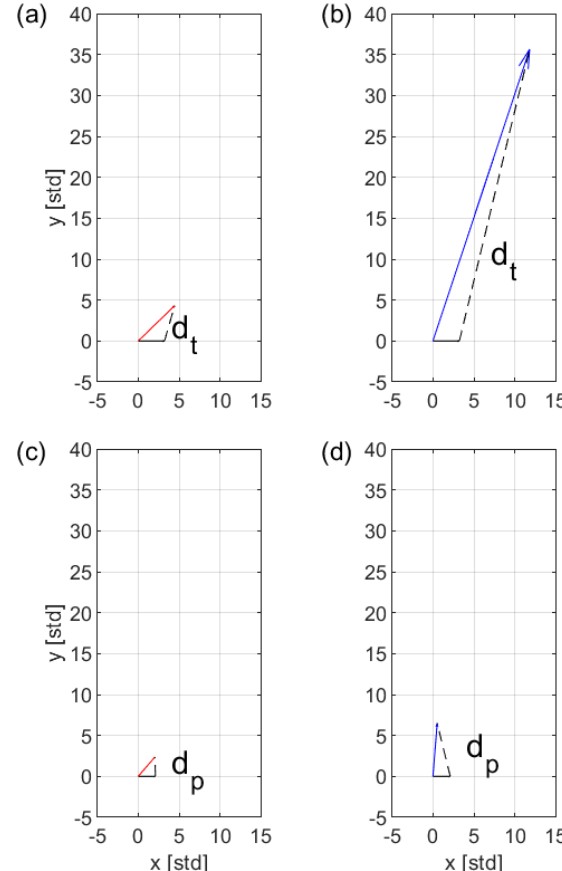

**Figure 7.** (a) The temperature distance between GLENS (black) and EQ (red), (b) the temperature distance between GLENS (black) and RCP8.5 (blue), (c) the precipitation distance between GLENS (black) and EQ (red), and (d) the precipitation distance between GLENS (black) and RCP8.5 (blue), shown on the 2D plane that contains both area-weighted vectors (while both vectors are $m$-dimensional, there is a unique plane that contains both). Both temperature and precipitation are expressed in number of standard deviations (and are thus dimensionless). In (a) and (b), std is the standard deviation of 20-year averages of temperature, calculated from the across-ensemble variability from 2010-2029 in RCP8.5 simulations. In (c) and (d), std is the standard deviation of 20-year averages of precipitation.

To estimate the relationships between $\theta_{AOD}$ and $d_t$ and between $\theta_{AOD}$ and $d_p$, we perform linear regressions on the data points obtained from the 10 pairwise comparisons among the 5 different SAI strategies in Table 1 (Fig. 8). We constrain the linear regressions to go through zero because an identical AOD pattern should yield an identical temperature and precipitation response. The linear functions are obtained as:





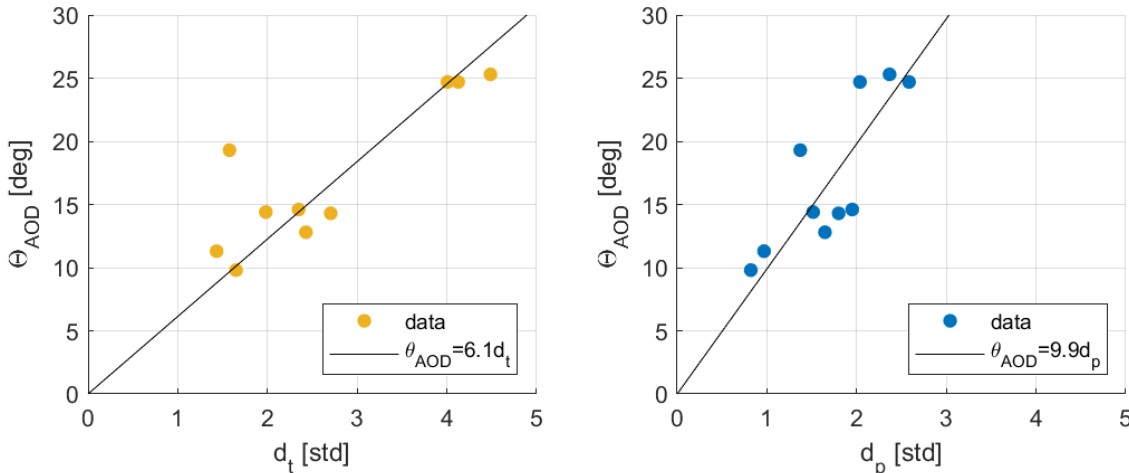

**Figure 8.** (a) Angle between AOD patterns, $\theta_{AOD}$, and the distance between corresponding temperature responses, $d_t$, for each pair of strategies in Table 1; (b) angle between AOD patterns, $\theta_{AOD}$, and the distance between corresponding precipitation responses, $d_p$, for each pair of strategies in Table 1. $d_t$ and $d_p$ are expressed in number of standard deviations (std) of 20-year averages of temperature and precipitation respectively. Orange dots represent the values of $\theta_{AOD}$ and corresponding $d_t$ and blue dots represent the values of $\theta_{AOD}$ and corresponding $d_p$ of all pairwise comparisons between different strategies. The black lines represent the best-fit linear regression functions, constrained to pass through the origin: $\theta_{AOD}$ =6.1 $d_t$, and $\theta_{AOD}$ =9.9 $d_p$ with the coefficient of determination, $R_t^2$ = 0.62 and $R_p^2$ = 0.65, respectively. The error in estimating each point (calculated from across-ensemble variability) is small (less than $0.2°$ in AOD angle, and less than 0.1 std in both temperature distance and precipitation distance) compared to the fitting error, indicating that a linear approximation to the relationship is only an approximation. Points in the upper-right (most dissimilar AOD and dissimilar surface climate) are the pairwise comparisons between the equatorial injection strategy and the other strategies. In (a), the outlier at approximately (1.6,19), which is the comparison between iSpring and iAutumn, shows that the relationship between $\theta_{AOD}$ and $d_t$ isn't exactly linear; similar AOD patterns yield similar climate responses but different AOD patterns do not guarantee different climate responses.

$$\theta_{AOD} = 6.1 d_t \tag{8}$$

$$\theta_{AOD} = 9.9 d_p \tag{9}$$

As shown in Fig. 8, pairs of strategies with relatively similar AOD patterns have relatively similar temperature and precipi-
tation responses, and conversely, pairs of strategies with very different AOD patterns result in very different temperature and precipitation responses.

Data used here are from SAI designs that were considered in previous studies. Although they are not designed to span either the overall AOD design space or the surface climate design space, these available simulations do provide a useful set of data for analyzing the relationship between how similar or dissimilar the AOD patterns are and how similar or dissimilar the surface
climate responses are.



# 5 Detectability at different levels of cooling

To evaluate how different the surface climate responses are, we first perform Welch's t-test on the five injection strategies, using a single ensemble member of each injection strategy. Welch's t-test assumes that sampled data are independent; we remove the effect of serial autocorrelation from the temperature and precipitation data by estimating the effective sample size assuming

both temperature and precipitation follow a first order autoregressive (AR(1)) process (Wilks, 2019). The t-test results for comparing differences in surface air temperature between GLENS and iSpring and between GLENS and EQ are shown in Fig. 9(a)-(b); the t-test results for precipitation comparison are shown in Fig. 10(a)-(b). At a 4°C cooling and a confidence level of 95%, temperature and precipitation responses from GLENS and iSpring are statistically significantly different from each other in 17% and 7% of Earth's area, respectively, as opposed to the comparison between GLENS and EQ, in which 37% and

23% of area on the Earth shows statistically significant difference in temperature and precipitation.

    Here, we define that two strategies are considered to be detectably different if the difference in temperature or precipitation responses between them are detectable at a 95% confidence level over a 20-year period on more than 5% of Earth's area. With the temperature and precipitation normalized by the standard deviation of 20-year means, the difference between the temperature or precipitation responses at any grid point will be detectably different if the difference between the normalized

data is more than 2. To obtain a global aggregate metric, we note that roughly 5% of the Earth's surface area has a temperature difference more than double the overall temperature distance $d_t$ considered earlier and 5% of the Earth's surface area has a precipitation difference more than double the precipitation distance $d_p$. (For example, between GLENS and iSpring, only 5.2% of Earth's area have a difference in regional temperature responses that is more than twice the value of $d_t$, and only 4.7% of Earth's area have a difference in regional precipitation responses that is more than twice the value of $d_p$.) Thus when the

temperature distance $d_t$ between two injection strategies is one standard deviation, then roughly 5% of the Earth's surface area will have detectably different temperature responses at a 95% confidence interval. Similarly, when the precipitation distance $d_p$ between two injection strategies is one standard deviation, then roughly 5% of the Earth's surface area will have detectably different precipitation responses at a 95% confidence interval. Thus, we use one standard deviation of the overall root mean square (RMS) normalized temperature distance or precipitation distance as the threshold for determining whether two strategies

result in detectably different temperature or precipitation responses.

    We compare the difference in temperature and precipitation responses between GLENS and iSpring and between GLENS and EQ, and show how the difference changes with levels of cooling using detectability plots, as shown in Fig. 9(c)-(d) and Fig. 10(c)-(d). Figure 9(c) and 10(c) show the detectability of difference in temperature and precipitation between GLENS and iSpring at a cooling level of 4°C. Figure 9(d) and 10(d) show the comparison between GLENS and EQ at the same

cooling level. In each plot, the length of the vector is equal to the area-weighted $L^2$-norm of the corresponding temperature or precipitation vector, and the distance between the two vectors is the corresponding temperature distance or precipitation distance. The circle around the tip of each vector represents the temperature or precipitation variability on a 20-year timescale, whose radius is equal to one due to the normalization by the standard deviation of temperature or precipitation variability. In Fig. 9(c), the two temperature variability circles are partially overlapped, while in Fig. 9(d), the two circles are separated from





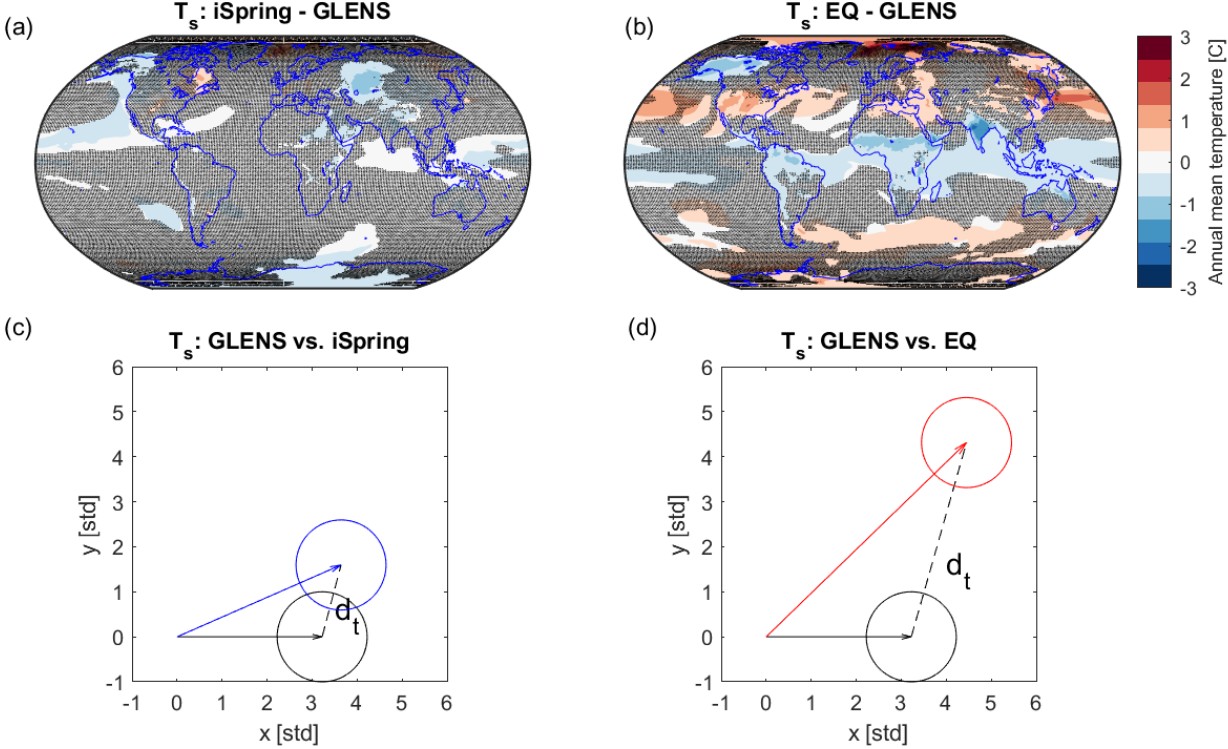

**Figure 9.** Left column shows the comparison between GLENS and iSpring; the right column shows the comparison between GLENS and EQ. (a) and (b) show the difference in temperature response between GLENS and iSpring and between GLENS and EQ, respectively; shaded areas are where no statistically significant difference is observed, based on a t-test with a confidence level of 95%. (c) and (d) are detectability plots that geometrically show the ability of detecting differences in the temperature responses between GLENS (black) and iSpring (blue) and between GLENS and EQ (red), respectively. The temperature responses are expressed in number of standard deviations ($\mathrm{std}$) of 20-year averages of temperature. The length of each vector represents the area-weighted $\mathrm{L}^2$-norm of surface air temperature response, and the circle represents temperature variability in all possible directions, with a radius of $1\ \mathrm{std}$ due to the normalization. The dashed line in (c) and (d) measures the temperature distance, $d_t$, between GLENS and iSpring and between GLENS and EQ, respectively. GLENS and EQ have a larger distance in the temperature response, which indicates the difference between GLENS and EQ is more detectable at the same level of cooling.

each other. The more overlapped these two circles have, the less area has detectably different temperature responses. When the overlapped area is sufficiently large, it indicates that the difference in the temperature responses is small enough such that it is hard to tell whether the difference could be purely due to natural variability. These implications from the observation of temperature responses also apply to precipitation.

From the detectability plots that compare these different SAI strategies, it is clear that the detectability of different injection

strategies depends on both the level of cooling and the choice of climate variables. With the underlying assumption of linearity





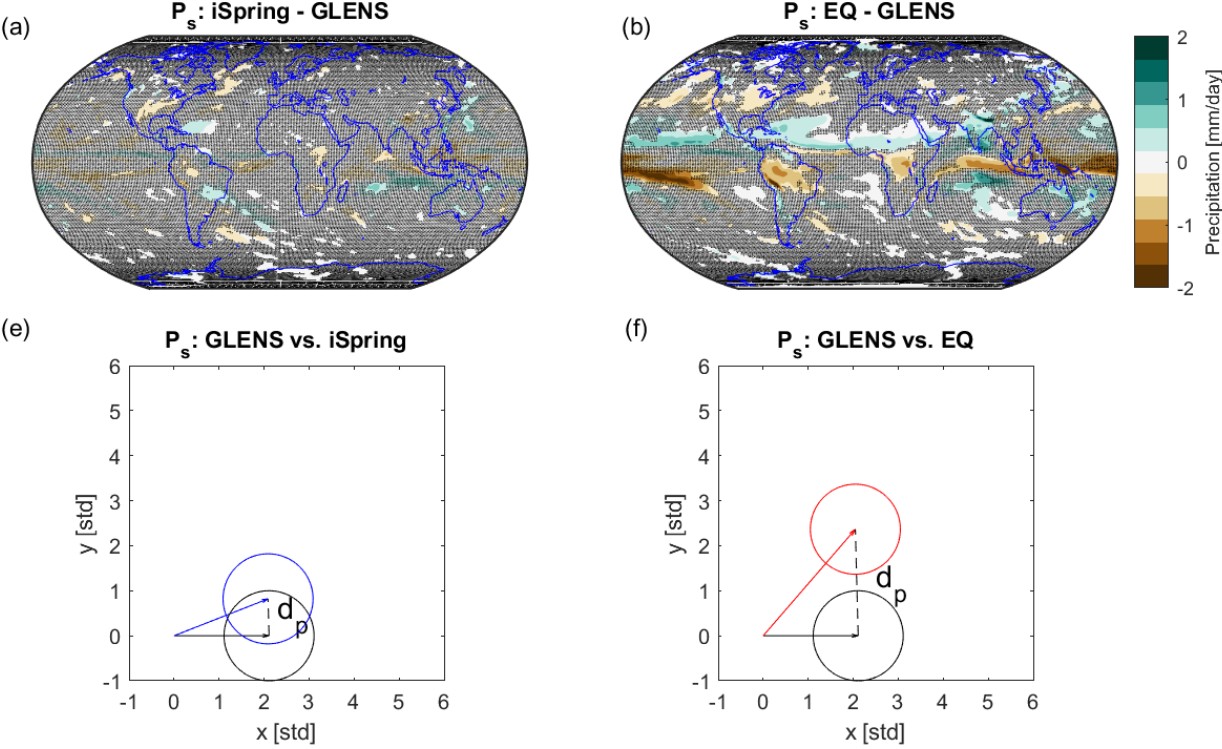

**Figure 10.** Left column shows the comparison between GLENS and iSpring; the right column shows the comparison between GLENS and EQ. (a) and (b) show the difference in precipitation responses; shaded areas are where no statistically significant difference is observed, based on a t-test with a confidence level of 95%. (c) and (d) are detectability plots that geometrically show the ability of detecting differences in the precipitation responses between GLENS (black) and iSpring (blue) and between GLENS and EQ (red), respectively. The precipitation responses are expressed in number of standard deviations (std) of 20-year averages of precipitation. The length of each vector represents the area-weighted $L^2$-norm of precipitation response, and the circle represents precipitation variability in all possible directions, with a radius of 1 std due to the normalization. The dashed line in (c) and (d) measures the precipitation distance, $d_p$, between GLENS and iSpring and between GLENS and EQ, respectively. Similar to temperature response, the difference in precipitation between GLENS and EQ is more detectable than that between GLENS and iSpring at the same level of cooling.

for surface climate responses, we estimate the difference in temperature and precipitation responses between GLENS and EQ at reduced levels of cooling (Fig. 11). For the same pair of strategies, as we reduce the amount of cooling, the temperature distance and the precipitation distance between these two strategies decreases. At 1.8°C cooling, the resulting temperature responses of these two strategies are 2 standard deviations of temperature variability away from each other (Fig. 11(b)); at the same level of cooling, the resulting precipitation responses are 1 standard deviation apart (Fig. 11(e)). At 0.9°C cooling, the resulting temperature responses are exactly 1 standard deviation apart (Fig. 11(c)), and the precipitation responses are 0.5 standard deviation apart (Fig. 11(f)). For any cooling level lower than 0.9°C, temperature responses of GLENS and EQ






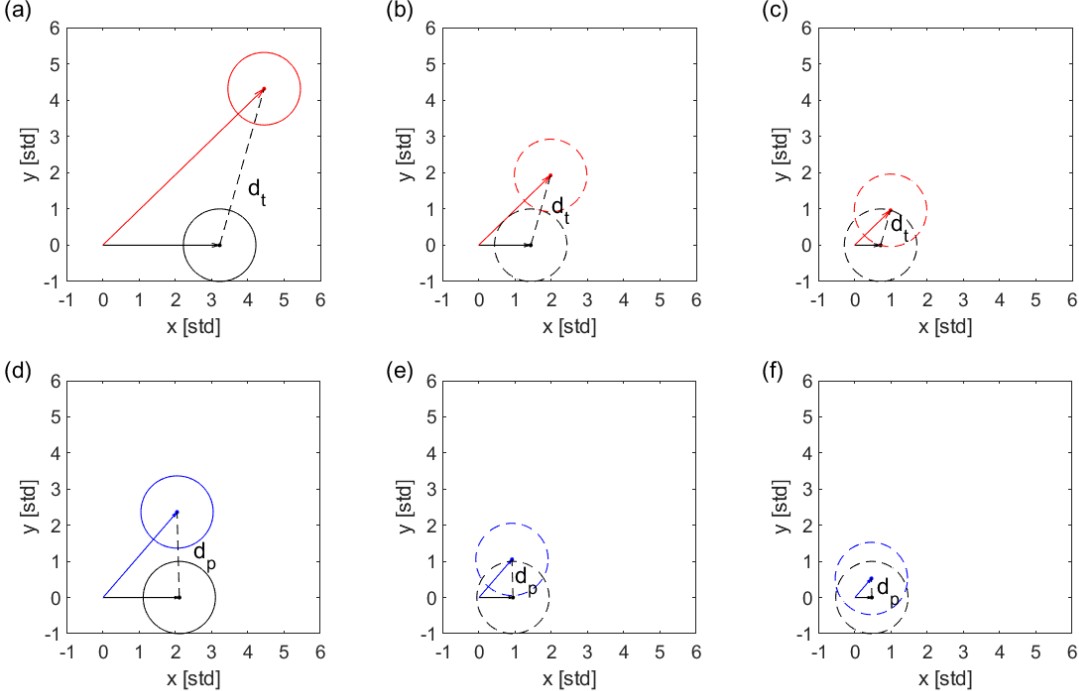

**Figure 11.** Top panels are detectability plots for the difference in temperature responses between GLENS (black) and EQ (red) at three different levels of cooling: (a) 4°C, (b) 1.8°C, and (c) 0.9°C. Bottom panels are detectability plots for the difference in precipitation responses between GLENS (black) and EQ (red) at those three levels of cooling. Dots represent the estimated mean temperature or precipitation responses for the corresponding levels of cooling. Temperature and precipitation responses are expressed in number of standard deviations (std) of 20-year averages of temperature and precipitation respectively. As the cooling level decreases, less area on the Earth has detectably different climate responses and the ability to detect the difference between GLENS and EQ decreases.

will not be detectably different by our metric; that is to say, less than 5% of area on the Earth is expected to have detectably different temperature responses at a 95% confidence level over a 20-year period. On the other hand, to have detectably different

precipitation responses between GLENS and EQ, the cooling level needs to be higher than 1.8°C. The cooling levels of 0.9°C and 1.8°C are defined as the cut-off cooling levels for detectable difference in temperature and precipitation, respectively, between GLENS and EQ. Note that for sufficiently small amounts of cooling, the resulting climate from these strategies will also be undetectably different from the climate with neither increased greenhouse gases nor SAI; from Fig. 7, they will be detectably different from the climate with increased GHG except at very small levels of cooling.





As shown in Fig. 11, the cut-off level of cooling $\Delta T_t$ is inversely proportional to $d_t$ and the cut-off level of cooling $\Delta T_p$ is inversely proportional to $d_p$:

$$\Delta T_t = \frac{4}{d_t} \tag{10}$$

$$\Delta T_p = \frac{4}{d_p} \tag{11}$$

By substituting eq. (10) into eq. (8) and substituting eq. (11) into eq. (9), we obtain two functions that can be used to estimate

a threshold value of AOD angle, $\theta_a$, which is used to assess the detectability of different injection choices at different levels of cooling:

$$\theta_a^t = 24/\Delta T_t \tag{12}$$

$$\theta_a^p = 40/\Delta T_p \tag{13}$$

Thus, given a particular level of cooling, we can calculate if two injection strategies are expected to result in detectably

different temperature or precipitation responses by comparing their AOD patterns. If the angle between the patterns of AOD is smaller than $\theta_a^t$ or $\theta_a^p$, these two strategies can be expected to not result in detectably different temperature or precipitation responses. However, if the angle between the patterns of AOD is larger than $\theta_a^t$ or $\theta_a^p$, these two strategies can be expected to be meaningfully independent in terms of temperature responses or precipitation responses. In Fig. 12, we compare the cut-off AOD angles predicted by eq. (12) and eq. (13). As shown in Fig. 12, the threshold values of AOD angle predicted using

temperature responses are always lower than those predicted using precipitation responses.

## 6   Estimating the Number of DOF

In the previous sections, we estimate the relationship between the number of DOFs included and the maximum error in approximating AOD, and the relationship between the AOD angles and the level of cooling at which the resulting temperature or precipitation response could be expected to be detectably different. In this section, we combine these two to estimate how

many meaningfully-independent DOF there are as a function of the levels of cooling.

As changes in temperature are more detectable than those in precipitation, the extent to which two AOD patterns are sufficiently similar, and thus the number of DOF in the SAI design space, are determined by the temperature response. Using eq. (12), we calculate the cut-off AOD angle $\theta_a^t$ (listed in Table 2) for cooling levels of 0.5°C, 1°C, 1.5°C, and 2°C. It is expected that two SAI strategies with AOD differing by any angle smaller than $\theta_a^t$, will not result in detectably different tem-

perature or precipitation responses. Figure 13 shows the four cut-off AOD angles and the corresponding minimum numbers of DOF required for $\theta^*(n)$ not exceeding the cut-off value, $\theta_a^t$ (also listed in Table 2).

$$\theta^*(n) \leq \theta_a^t(\Delta T) \tag{14}$$

Any set of injection choices that has a $\theta_{max}$ smaller than the cut-off angle $\theta_a^t(\Delta T)$ will form a design space that captures all detectably different climate responses. That is, for any possible injection strategy not included in the design space, you can find





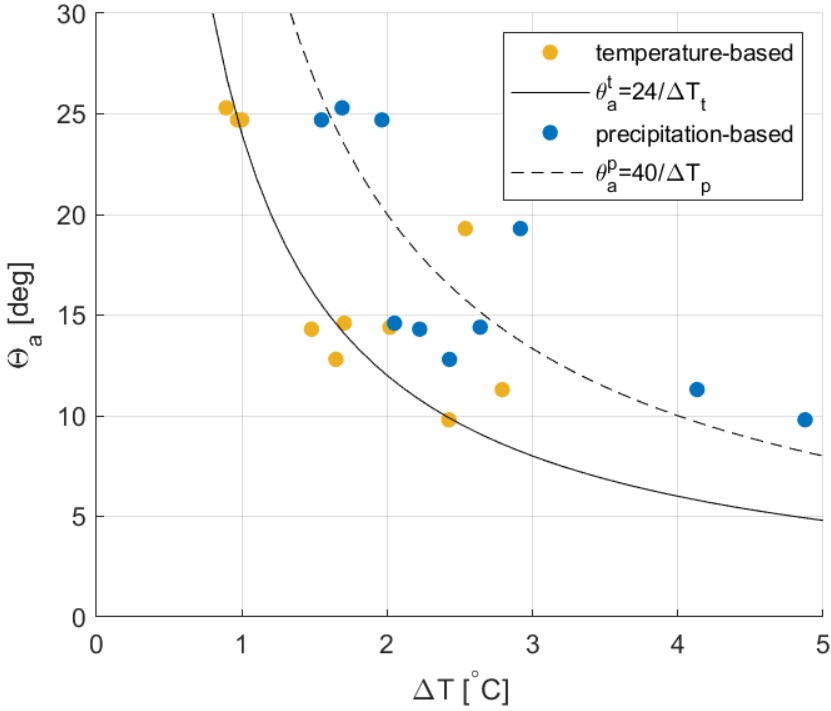

**Figure 12.** The cut-off AOD angle $\theta_a$ at different levels of cooling $\Delta T$ estimated using temperature responses and precipitation responses.

**Table 2.** The minimum number of DOF, $n$, of the SAI design space for four targeted levels of cooling. At each level of cooling, $\theta_a^t$ is the maximum angle that can be formed between two AOD patterns that yield undetectably different temperature responses. $\theta^*(n)$ is the maximum angle that can be formed between any AOD pattern and the design space of $n$ DOF, and must be smaller than $\theta_a^t$.

| Targeted level of cooling [$^\circ C$] | Cut-off angle $\theta_a^t$ | Minimum number of DOF, $n$ | $\theta^*(n)$ |
|---|---|---|---|
| 0.5 | 48° | 2 | 38° |
| 1 | 24° | 6 | 18° |
| 1.5 | 16° | 8 | 15° |
| 2 | 12° | 12 | 11° |

an injection strategy in the design space such that the angle between their AOD patterns is smaller than $\theta_a^t$ and the difference in the corresponding climate responses is sufficiently small such that they are not meaningfully different.

As shown in Table 2, as the targeted level of cooling increases, the cut-off value of $\theta_a^t$ decreases and the minimum number of DOF increases. For a targeted cooling level of 1°C, we likely need 6 DOFs. For any possible AOD pattern, the angle between it and the AOD pattern approximated by a design space of 6 DOFs is likely to be smaller than 24° , and the resulting difference

in climate response will be largely undetectable. For a targeted cooling level of 1.5°C, we likely need 8 DOFs. This finding

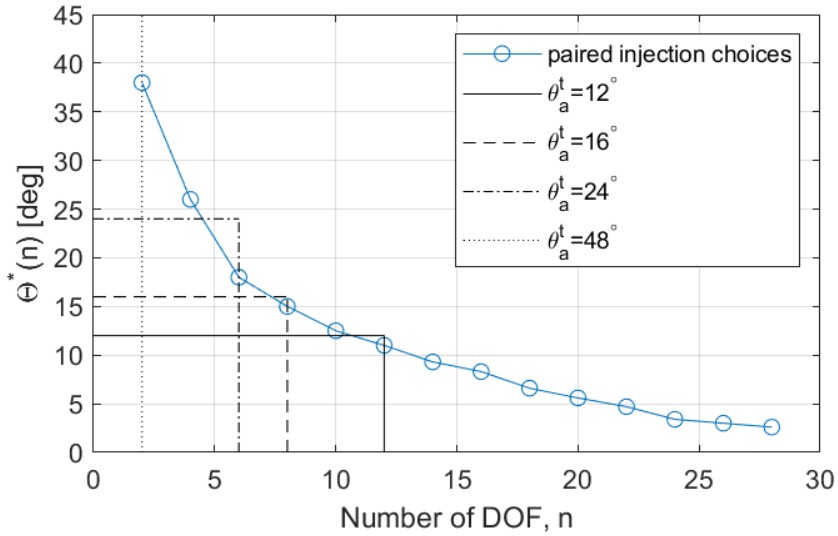

**Figure 13.** The minimum number of DOF corresponding to the worst-case error in approximating AOD, $\theta_a^t$.

significantly reduces the dimension of the design space needed for evaluating the possible climate impacts of different SAI strategies and associated trade-offs.

For a cooling level of 1°C, the best set of 6 among the set of 29 sample injection choices are: (i) spring injections at 60° N/60° S, (ii) annual injections at 30° N/30° S and (iii) winter and summer injections at the equator. Figure 14 shows the AOD

spatiotemporal patterns of the six injections in the best set. A set of 6 that instead includes annually-constant injection at 30° N, 15° N, 15° S, and 30° S (the four cases considered in MacMartin et al. (2017), Kravitz et al. (2017), and Tilmes et al. (2018)) as well as spring injection at 60° N (as in Lee et al. (2021)) and 60° S is only slightly worse than this optimal set, within a range of 0.1° , still sufficient to span the design space for 1°C cooling.

For a cooling level of 1.5°C, the best set of 8 is similar to the best set of 6 but with equatorial injections at the other two

seasons instead and the addition of summer injections at 15° N and 15° S. Note that in the optimization earlier, we constrained our search to hemispherically symmetric pairs of injection choices. Including only annually-constant injection at the equator rather than two seasons yields a set of 7 injection choices that performs almost as well as the optimal set of 8 and is still sufficient for 1.5°C cooling.

## 7   Analysis of injections at different altitudes

The SAI simulations analyzed in the previous sections are all high-altitude injections (6-7 km above the tropopause). Tilmes et al. (2017) also conducted low-altitude (5 km lower) simulations at 50° N, 30° N, 15° N, 0°, 15° S, 30° S and 50° S with an annual injection rate of 6 Tg yr$^{-1}$, all simulated in CESM1(WACCM). The AOD patterns of low-altitude and high-altitude injections are shown in Fig. 15. For each injection case, the spatiotemporal AOD responses are weighted by TOA incoming





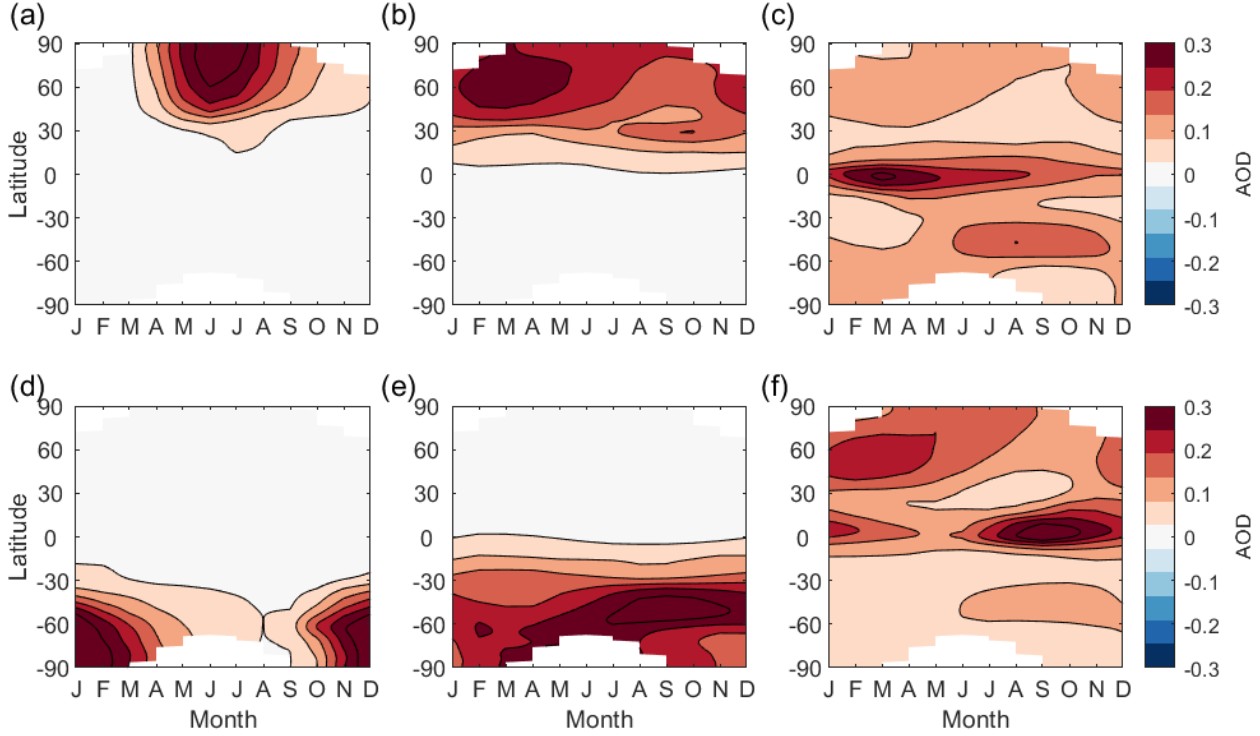

**Figure 14.** AOD patterns from the best set of 6 injections: (a) spring injection at 60° N, (b) annual injection at 30° N, (c) winter injection at 0° (d) spring injection at 60° S, (e) annual injection at 30° S, and (f) summer injection at 0°.

**Table 3.** Angle between the AOD vector of each high-altitude injection and the set of all low-altitude injections.

| Injection latitude | 50° N | 30° N | 15° N | 0° | 15° S | 30° S | 50° S |
|---|---|---|---|---|---|---|---|
| $\theta_{AOD}$ | 5.3° | 11.6° | 6.2° | 7.0° | 4.5° | 6.3° | 4.9° |

solar energy. The angle of the AOD pattern of each high-altitude injection with respect to the set of all low-altitude injections is
listed in Table 3; these high-altitude injections are all within a small angle with respect to the set of low-altitude injections. For
a level of cooling under 2°C, the difference of AOD responses due to injecting at these different altitudes is small compared
to the differences achievable through injecting at different latitudes and seasons. If a much higher level of cooling is desired,
injecting at different altitudes may result in meaningfully different surface climates and injection choices of different altitudes
may need to be considered when choosing the design space and evaluating trade-offs. Injection at or below the tropopause,
while inefficient, would likely result in more significant differences in AOD patterns (Bernstein et al., 2013).




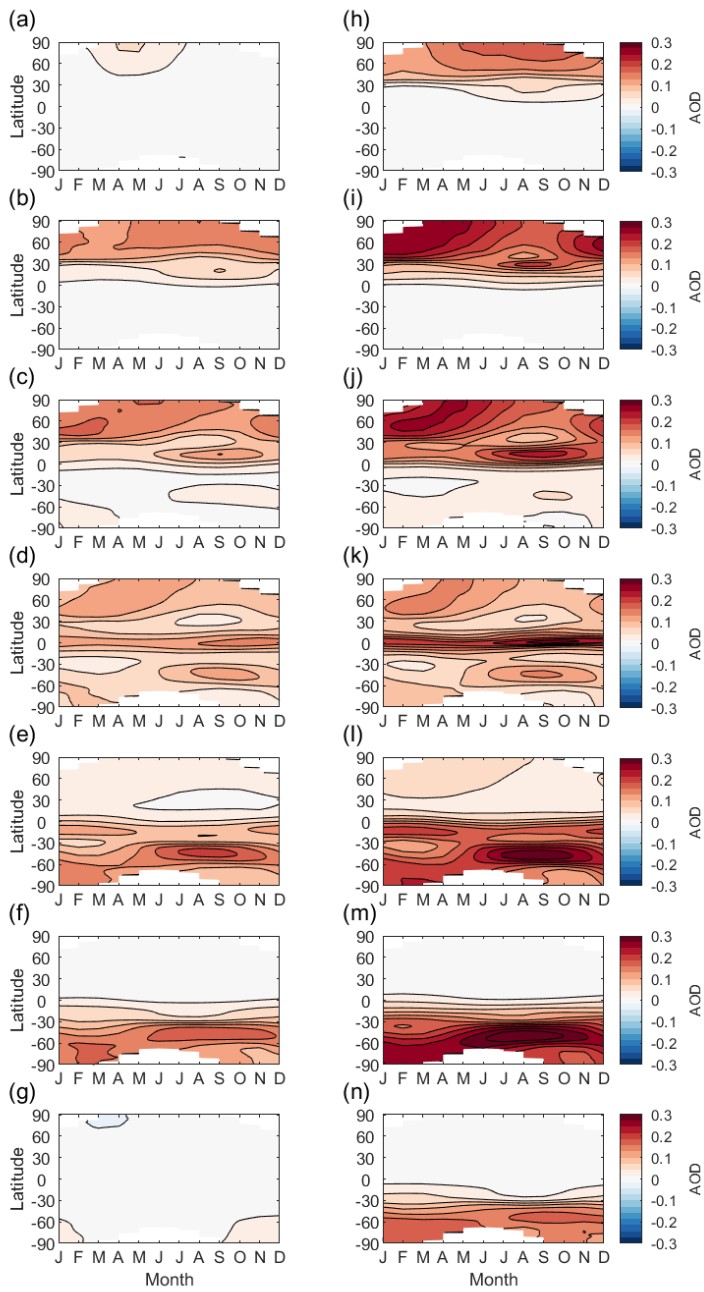

**Figure 15.** AOD patterns produced by low-altitude annually-constant injection of 6 Tg yr$^{-1}$ at (a) 50° N, (b) 30° N, (c) 15° N, (d) 0°, (e) 15° S, (f) 30° S, (g) 50° S and high-altitude annually-constant injection of 6 Tg yr$^{-1}$ at (h) 50° N, (i) 30° N, (j) 15° N, (k) 0°, (l) 15° S, (m) 30° S, (n) 50° S.





# 8    Conclusions

Previous studies have shown that different choices of stratospheric aerosol injection latitudes and seasons lead to different surface climate responses. Choosing where and when to inject aerosols to the stratosphere to meet different climate goals can be considered as a design problem. Previous studies have concluded that there are at least 3 degrees of freedom (DOF), that is,

at least three independent climate goals can be simultaneously met. These three – basically the global mean aerosol burden, the interhemispheric difference, and the equator-to-pole difference – were motivated by physical intuition regarding stratospheric transport, which will ultimately constrain how many independent DOFs are achievable through different injection choices. A key observation is that the number of DOF effectively depends on the amount of global cooling provided by SAI, because for a small amount of cooling, the difference in the climate response for different strategies may not be detectable. As the amount

of cooling increases, the number of meaningfully-independent DOF increases. For a cooling level of 1-1.5°C, there are likely between 6 and 8 meaningfully-independent DOFs. If only precipitation changes mattered, and not temperature changes, then there would be fewer meaningfully-independent DOFs.

Our estimation of the number of DOF provides useful guidance to bound the number of injection choices that need to be considered when evaluating the range of possible different SAI strategies and the trade-offs among them. If only a small

amount of cooling is needed from implementing stratospheric aerosol injections, a small set of selected injection choices would be sufficient to capture the range of possible resulting climate responses and evaluate how different those climate responses could be. As all possible injection choices form an extremely high dimensional design space, only considering the meaningfully-independent injection choices significantly reduces the dimension of the design space.

The number of meaningfully-independent DOF determines the number of independent climate metrics that SAI can manage

simultaneously. Thus, for a cooling level of 1-1.5°C, for example, SAI can manage 6-8 independent climate metrics at the same time. This expands the manageable number of climate metrics relative to what has been considered in previous studies, opening up new opportunities for exploring alternate designs that will have different distributions of impacts.

It is important to note that all of these results are obtained from a single climate model. Other climate models may produce different numerical results. Nonetheless, the number of independent DOF needed to span the range of possible different

stratospheric AOD patterns can be reasonably expected to remain consistent as the transport of aerosols are constrained by stratospheric circulation. We make several simplifying approximations in order to make analysis tractable, particularly to estimate the relationship between how similar or dissimilar two patterns of AOD are and how similar or dissimilar the corresponding surface climate is; future research could explore the impact of these approximations. First, we only consider changes in temperature and precipitation, and we only consider changes in annual mean rather than shifts in seasonality, which could

matter at high latitudes in particular (Jiang et al., 2019). Second, we only consider whether the difference in climate response would be detectable over a 20-year period. Third, the globally-aggregated metric we use for determining whether two different climate responses are "detectable" is based on whether they are detectably-different at a 95% confidence level over 5% of the surface area. However, changes that are less-confidently detected may still matter, and since social, agricultural and other economic activities are strongly affected by regional climate changes, just because they only happen in a small percentage



of area, does not necessarily mean that they are not important – the details of where and what the differences are potentially matter.

A key outcome of this study is that further research should be conducted to explore alternate SAI designs that can manage more than 3 and up to 8 independent climate metrics simultaneously, and to compare the resulting climate responses and associated trade-offs. Research into more than 8 is less policy-relevant, simply because any hypothetical deployment scenario

would not reach more than 1.5 °C cooling for many decades, if ever. Ultimately, to evaluate the impacts of stratospheric aerosol geoengineering, regional surface climate needs to be considered, as social and economic activities are significantly affected by regional climate change. Alternate SAI designs may enable a better compensation of the impacts from climate change; alternate designs might also lead to the potential to create more novel climates that optimize some metrics at the expense of others – both of these possibilities are important to understand in order to inform not only future scientific research in SAI but

also governance challenges.

*Data availability.* Data for the new simulations presented in this study (specifically, monthly aerosol optical depth (AOD) for spring injections at 60°N, 45°N, 45°S, and 60°S and annually-constant injections at 7.5°N, 22.5°N, and 37.5°N) are available through the Cornell e-Commons Library at https://doi.org/10.7298/f1e4-sq40 (Zhang et al., 2021). Data for GLENS and equatorial (from Tilmes et al., 2018 and Kravitz et al., 2019 respectively) are available at https://doi.org/10.5065/D6JH3JXX, data for iSpring and iAutumn (from Visioni et al.,

2020c) are available at https://doi.org/10.7298/c92j-2p46 (Visioni et al., 2020a), and data for PREC (from Lee et al., 2020a) are available at https://doi.org/10.7298/d2qm-1568 (Lee et al., 2020b).

*Author contributions.* YZ conducted all analyses and wrote the paper with editing from DM, BK and DV; YZ and DM conceived the study with input from all authors and DV assisted with conducting simulations.

*Competing interests.* The authors declare that they have no conflict of interest.

*Acknowledgements.* The authors would like to acknowledge high-performance computing support from Cheyenne (https://doi.org/10.5065/D6RX99HX) provided by NCAR's Computational and Information Systems Laboratory, sponsored by the National Science Foundation. Support for Y. Zhang and D. G. MacMartin was provided by the National Science Foundation through agreement CBET-1818759 and CBET-2038246. Support for D. Visioni was provided by the Atkinson Center for a Sustainable Future at Cornell University. Support for BK was provided in part by the National Science Foundation through agreement CBET-1931641, the Indiana University Environmental Resilience Institute,

and the Prepared for Environmental Change Grand Challenge initiative. The Pacific Northwest National Laboratory is operated for the U.S. Department of Energy by Battelle Memorial Institute under contract DEAC05-76RL01830. The CESM project is supported primarily by the National Science Foundation.



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
