# Peer review of "How large is the design space for stratospheric aerosol geoengineering?"

_Earth System Dynamics, 2021_

## Author Response (AR1)

**Author responses to referee comments (RCs) on "How large is the design space for stratospheric aerosol geoengineering".**

Yan Zhang, Douglas G. MacMartin, Daniele Visioni, Ben Kravitz

**Referee #1:**

*Original referee comments are in italics*
Author responses are in plain text

*The submitted paper discusses the number of independent degrees of freedom in climatic response to stratospheric sulfate injection as a mechanism for geoengineering. The paper considers a superset of existing simulations using CESM, where aerosol injections are varied by latitude and season. The authors identify an approach for assessing the similarity of both the resulting Aerosol Optical Depth (AOD) pattern and the resulting perturbations in regional temperature and precipitation. Independent degrees of freedom are isolated by considering both the optimal/minimal subset of modes to linearly reconstruct the remaining ensemble of forcing patterns, and the resulting detectability of reconstruction errors in the context of internal variability.*

*The paper is well written and comprehensive, and addresses an interesting and pertinent point with respect to geoengineering strategy design. The mathematical formulation is a little unconventional, and the results are more empirical than might be strictly necessary. The authors should also take care to avoid statements that imply that their design space is an inherent property of the system (rather than a subset of the small number of injection experiments conducted to date). That said, the paper makes some strong and interesting points is quite adequate for publication in ESD with just some minor clarifications.*

We thank the referee for their comments on our paper, and respond individually to each of their comments below.

*Minor points*

*1 - The approach is semi-empirical, and subject to the properties of the parent ensemble. Though the end product is useful, the design space is only a subset of the possible degrees of freedom in the response to injection. Several additional degrees of freedom are unexplored in the parent ensemble - longitudinal variation in injection site, additional injection altitudes, adaptive injection (i.e. responding to synoptic conditions), variation in background scenario - which could potentially increase the design space. This should be highlighted more clearly.*

We thank the referee for their comment on the suggestion on additional degrees of freedom that could potentially increase the design space. We have better highlighted the general observation made by the reviewer in our revision. In the "Introduction" section, L32, we added "Here, we only consider possible injection choices at a finite set of different latitudes and seasons. We

validate in Section 7 that different choices for injection altitude do not produce meaningfully-independent patterns of AOD. The injection longitude would not be expected to matter due to the rapid zonal mixing relative to the aerosol lifetime". In the "Conclusions" section, L413, we added "When evaluating the design space, we do not consider injections at different longitudes and additional altitudes beyond those evaluated in Section 3, nor do we consider adaptive injection strategies as explored by Aksamit et al (2021). The longitude is not expected to matter due to rapid zonal mixing.  As the injection altitude approaches the tropopause, the lifetime in the stratosphere decreases, leading to more spatially- and temporally-confined changes to AOD that would add further degrees of freedom at the expense of a significant loss of efficiency". Here, we explain below why we did not consider the other degrees of freedoms in our paper:

(1) The stratospheric circulation is dominated by meridional circulation (Brewer-Dobson Circulation) and isentropic mixing. Due to strong zonal wind, aerosols in the stratosphere are mixed very quickly in the longitudinal direction. Aerosol formation and zonal mixing takes about the same amount of time. Thus, the injection longitude would not be expected to matter, especially for the range of global cooling considered in this study.

(2) In our "Section 7 Analysis of injections at different altitudes", we do explore the effect of injecting at different altitudes and compare the patterns of AOD from injections at two different altitudes. As the injection altitude gets closer to or below the tropopause, the lifetime of stratospheric aerosols reduces, and the patterns of AOD become more spatially localized at the expense of a significant loss of injection efficiency, and this could add further degrees of freedom; we now note this in discussing the role of altitude in Section 7.

(3) Different background scenarios project different levels of global warming, and thus may desire different levels of cooling. As we conclude from our study, the number of degrees of freedom increases with the amount of cooling. A higher emission scenario that projects larger global warming might lead to larger desired cooling, which in turn would lead to more degrees of freedom being required, but the background scenario itself will have at most a very minor impact on the number of degrees of freedom needed to span the design space for a given cooling. In L440, we added "Larger cooling might be needed under a higher emission scenario, which would add further degrees of freedom".

(4) Time-varying injection locations that harness short-term stratospheric diffusion barriers is studied in Aksamit et al., 2021. The primary benefits are found over relatively short timescales compared with the aerosol lifetime in the stratosphere, and thus while it is possible that this adaptive strategy will add additional degrees of freedom to the design space, we do not expect it to be a significant limitation on our study. We have added a citation to Aksamit et al. (2021) in the paper.

*2 - The iterative approach of selecting subset members by their ability to represent the remaining ensemble through linear combination is logical - but this may still not reveal the minimum possible number of degrees of freedom, and is again subject to the original ensemble sampling. The patterns maximize independence within the source ensemble, but they are not orthogonal (as is strictly implied by the linear concept of a degree of freedom).  An orthogonal*

*set of patterns may be able to describe the ensemble variation of AOD forcing with a smaller number of degrees of freedom than those found in this study.*

We thank the referee for their comment on the selected sample set. Regarding the original ensemble sampling, our sample set includes injections at different latitudes from 60°N to 60°S with 15° resolution and in either the entire year or one of the four seasons. We demonstrate that this latitudinal resolution is adequate by showing that annual injections at 7.5°N, 22.5°N, and 37.5°N can be reasonably represented by a linear combination of the injections in the sample set. While this does not guarantee that there is no additional choice that was not considered, it does lend confidence that we have adequately sampled the set, since AOD patterns are constrained by stratospheric circulation. Although we do not look at latitudes higher than 60°, Lee et al.(2021) have pointed out in their paper that the AOD pattern produced by injection at 75°N is similar to the AOD pattern produced by injection at 60°N; we now note that explicitly in the revised paper. We thus think it is reasonable to expect that our selected sample set reasonably represents the space of possible AOD patterns for cooling levels not exceeding 2°C.

It is true that the patterns from a given subset of injection choices are not orthogonal, but since they are linearly independent, they could certainly be orthogonalized through a linear transformation; this does not change the number of degrees of freedom.

In L107, we added "In addition, Lee et al.(2021) have pointed out that the spatiotemporal distribution of AOD arising from injection at 75°N is similar to that arising from injection at 60°N and injecting at latitudes north of 75°N provides diminishing returns in terms of albedo enhancement; thus, injections at latitudes higher than 60° are not included in this sample set".

In L150, we added "Any given subset of linearly independent injection choices do not produce an orthogonal set, but could be orthogonalized if needed".

*3 - The choice of 20 year mean pattern differences as a metric for signal emergence may mask important aspects of the climate, especially for precipitation. Extreme frequency, drought, variability, seasonality - are all not represented in this mean state metric - and though this limitation is noted in the discussion, but it is potentially answer-changing in terms of the degrees of freedom in response. Precipitation response in particular to may have additional detectable degrees of freedom in the source ensemble if other climatological metrics are taken into account.*

We thank the referee for their comments. We agree that the choice of 20-year average, as well as the choice of 95% confidence threshold over 5% of the Earth area, affect the number of degrees of freedom that are "meaningfully-different". We have made this point more explicitly in the "Conclusions" section.
Surface temperature response is primarily controlled by thermodynamics, but precipitation is also affected by atmospheric circulation, which is controlled by dynamics. Current climate models have relatively high confidence in predicting temperature responses but have lower confidence in predicting circulation-related responses, and much lower confidence in predicting

regional-scale extreme weather events. While extreme weather events are critical in impact analysis, with the low confidence due to natural variability and model error, changes in the probability and severity of these events are typically harder to detect than changes in annual mean temperature or precipitation. Thus while it is reasonable to consider other metrics in future work, it is also reasonable to expect that the difference between any two strategies will be more readily detectable in annual-mean temperature than in different extreme events. Thus, including additional metrics is not likely to affect the number of degrees of freedom. We have explicitly noted this in the "Conclusions" section.

We added in the "Conclusions" section that "the choice of 20-year average, as well as the choice of 95% confidence threshold over 5% of the Earth area, affect the number of DOF that are "meaningfully-different". Considering the responses over a longer period of time might introduce additional DOF. To date, current climate models have relatively high confidence in predicting temperature responses but have lower confidence in predicting circulation-related responses, and much lower confidence in predicting regional-scale circulation-related extreme events (Shepherd, 2014). The number of DOF is primarily driven by the differences in temperature responses. Precipitation and circulation-related extreme events are typically harder to detect and are not likely to introduce additional DOF; that is, as the amount of cooling is gradually increased (or as the time horizon is increased), two distinct strategies are likely to become detectably different in their temperature response earlier than for precipitation changes or changes in extremes".

*4 - Aspects of the language regarding SAI in policy in the abstract and introduction need revision to highlight the uncertainty in the efficacy and risks associated with SAI. For example, the second line in the abstract should perhaps read: "Adding aerosols to the lower stratosphere has been modeled to produce temporary global cooling." The first sentence of the introduction is overly leading and unjustified. A cessation in GHG emissions today would prevent further warming. Policy may fail to deliver these reductions - but it is not the concept of emissions reductions itself which would be insufficient to prevent escalating climate risks. It should also be noted that SRM does not only have the capacity to reduce risks - it also has the potential to create additional risks relative to a non-SRM conventional mitigation scenario (e.g. rapid climate change in the event of early cessation).*

We thank the referee for the suggestion on modifying the language that we use to reflect the uncertainties and potential risks associated with SAI. We have updated the abstract and introduction to reflect these aspects.
We do not totally agree with the referee for the suggested change of the second sentence in the abstract. Existing observational data from volcanic eruptions have shown that adding aerosols to the stratosphere can temporarily reduce global mean temperature; that conclusion is thus not simply something found in models. We changed that sentence to "Adding aerosols to the lower stratosphere would result in temporary global cooling".
In the Introduction section, Line 15, we changed that sentence to "As a supplement to emission reduction, climate interventions such as stratospheric aerosol injection (SAI) may be able to temporarily reduce some of these risks".

*Technical points:*
*Subfigure 10 should be labeled (a-d)*

We have corrected the labelling in Figure 10.

**Author responses to referee comments (RCs) on "How large is the design space for stratospheric aerosol geoengineering".**

Yan Zhang, Douglas G. MacMartin, Daniele Visioni, Ben Kravitz

**Referee #2:**

*Original referee comments are in italics*
Author responses are in plain text

*The manuscript provides and demonstrates a mathematical method to quantify the degree of freedom of stratospheric aerosol injection (SAI) geoengineering. The authors transform the simulated AOD pattern and temperature and precipitation responses to the vectors in multi-dimensional spaces and explore their relationship, then use it to search the meaningfully-independent injection choices. As there are lots of possible injection choices in SAI geoengineering research, the manuscript will be useful to guide future SAI studies, although the sample space used in this study is limited due to available SAI simulations and all samples are based on a single earth system model. In general, the manuscript is well written and fits the scope of the ESD. However, the manuscript uses some specific mathematical tools, it is necessary to make the manuscript more readable for a wider geoscience community.*

We thank the referee for their comments on our paper, and respond individually to each of their comments below.

*Minor points:*

*1. As similar AOD patterns yield similar climate responses but different AOD patterns do not guarantee different climate responses, how does this might affect searching the meaningfully-independent injection choices?*

We thank the referee for this question. This means that one might explore too many options and the effective number of DOF might be smaller than what we found in our study (which is better than the converse problem of possibly ignoring important DOF). In L281, we have added explanations to make this point more clear.

*2. The AOD patterns are constrained by the stratospheric circulation and lifetime of the aerosol. Is it feasible to explore more possible AOD patterns by using widely available non-geoengineering CMIP simulations? With more AOD patterns, the sample space can be well expanded.*

We thank the referee for this suggestion. Using the non-geoengineering CMIP simulations would be a potentially valuable complement to our analysis. Available simulations include, for example, the response to volcanic eruptions at different latitudes, altitudes, and times of year. However, this dataset of opportunity may not fully span the design space. Compared to using

non-geoengineering data, our approach has two advantages: (1) Our sample set has good resolution of injection latitudes and seasons. The AOD patterns obtained from available non-geoengineering data depend on the location of active volcanoes and the time of volcanic eruptions, which may not have a better spatiotemporal resolution. (2) Our sample set allows a direct connection back to the injection choices that span the design space. Using the AOD from simulations of volcanic eruptions does not clearly reveal a direct relationship between a certain injection choice and the AOD pattern arising from a volcanic eruption.

*Technical points:*

*L34-35: Please clarify why the number of independent injection choices is equivalent to the number of independent climate goals.*

We assess meaningfully-independent injection choices based on whether different choices yield sufficiently distinct patterns of AOD that they can be expected to yield meaningfully different climate responses. Thus, the number of independent injection choices is equal to the number of meaningfully different climate responses. Assuming that the climate response can reasonably be approximated as linear, the ability to independently combine $n$ patterns of climate response allows $n$ independent goals to be managed, provided that they do not conflict.

We have modified the text to make this point clear.

*L136-139: Please clarify the definitions of the length of vector and the angle between the vectors.*

Thank you. We have modified those sentences to clarify the definition of the length of vector and the angle between the vectors.

*L261: Do the vectors of T and P responses adopt similar definition for the vectors of AOD pattern? If yes, it is better to explain them together.*

They adopt different definitions. The vectors of T and P include annual mean surface responses at different longitudes and different latitudes, while the vector of AOD includes monthly zonal mean AOD values.

*L263: Define the abbreviation "EQ" at its first appearance.*

Thank you. We have defined "EQ" in Table 1.

*L429: Extreme events can be part of future consideration due to their profound impacts on social and economic activities.*

We agree with the referee that extreme weather events should be considered in future research, although they are not likely to add further degrees of freedom for the range of cooling

considered in this paper. We have noted this in the "Conclusions" section, as noted in our response to the other reviewer.

*Figure 9, 10 and 11: Please explain their x & y axes.*

We have added explanations in the figure captions, that "both x- and y-axis are distances in number of standard deviations".